# KGOT: Unified Knowledge Graph and Optimal Transport Pseudo-Labeling for Molecule-Protein Interaction Prediction

**Jiayu Qin**[1,2]   **Zhengquan Luo**[2]   **Guy Tadmor**[3]   **Changyou Chen**[1]   **David Zeevi**[3]   **Zhiqiang Xu**[2]

[1]Department of Computer Science and Engineering, University at Buffalo, USA
[2]Mohamed bin Zayed University of Artificial Intelligence, UAE
[3]Weizmann Institute of Science, Israel
`{jiayuqin, changyou}@buffalo.edu`
`{zhengquan.luo, zhiqiang.xu}@mbzuai.ac.ae`
`{guy.tadmor, david.zeevi}@weizmann.ac.il`

## Abstract

Predicting molecule-protein interactions (MPIs) is a fundamental task in computational biology, with crucial applications in drug discovery and molecular function annotation. However, existing MPI models face two major challenges. First, the scarcity of labeled molecule-protein pairs significantly limits model performance, as available datasets capture only a small fraction of biologically relevant interactions. Second, most methods rely solely on molecular and protein features, ignoring broader biological context—such as genes, metabolic pathways, and functional annotations—that could provide essential complementary information. To address these limitations, our framework first aggregates diverse biological datasets, including molecular, protein, genes and pathway-level interactions, and then develops an optimal transport-based approach to generate high-quality pseudo-labels for unlabeled molecule-protein pairs, leveraging the underlying distribution of known interactions to guide label assignment. By treating pseudo-labeling as a mechanism for bridging disparate biological modalities, our approach enables the effective use of heterogeneous data to enhance MPI prediction. We evaluate our framework on multiple MPI datasets including virtual screening tasks and protein retrieval tasks, demonstrating substantial improvements over state-of-the-art methods in prediction accuracy and zero-shot ability across unseen interactions. Beyond MPI prediction, our approach provides a new paradigm for leveraging diverse biological data sources to tackle problems traditionally constrained by single- or bi-modal learning, paving the way for future advances in computational biology and drug discovery.

## 1 Introduction

Molecular and protein representation learning is an increasingly important topic in computational biology and drug discovery (Jumper et al., 2021; Zhou et al., 2023), fueled by the availability of large-scale *unlabeled* datasets (Consortium, 2024; Guo et al., 2025; AlQuraishi, 2019; Nakata et al., 2020). These resources have enabled the development of powerful molecular and protein encoders, which serve as foundational models for various downstream tasks. For example, self-supervised learning approaches (Rives et al., 2019; Zhou et al., 2023) leverage massive sequence and structure databases to capture intricate biochemical properties without requiring explicit supervision. This line of research not only advances standalone molecular/protein modeling but also lays the foundation for pushing the boundaries of computational biology and accelerating medicine and life science discoveries.

**What is new in KGOT?**   While prior work explores knowledge-enhanced drug–target prediction, existing approaches typically (i) treat knowledge graphs as an auxiliary feature source and train on observed edges, or (ii) apply pseudo-labeling heuristics that do not explicitly enforce global consistency. KGOT instead introduces a unified *Optimal Transportation(OT)+Knowledge Graph(KG)*

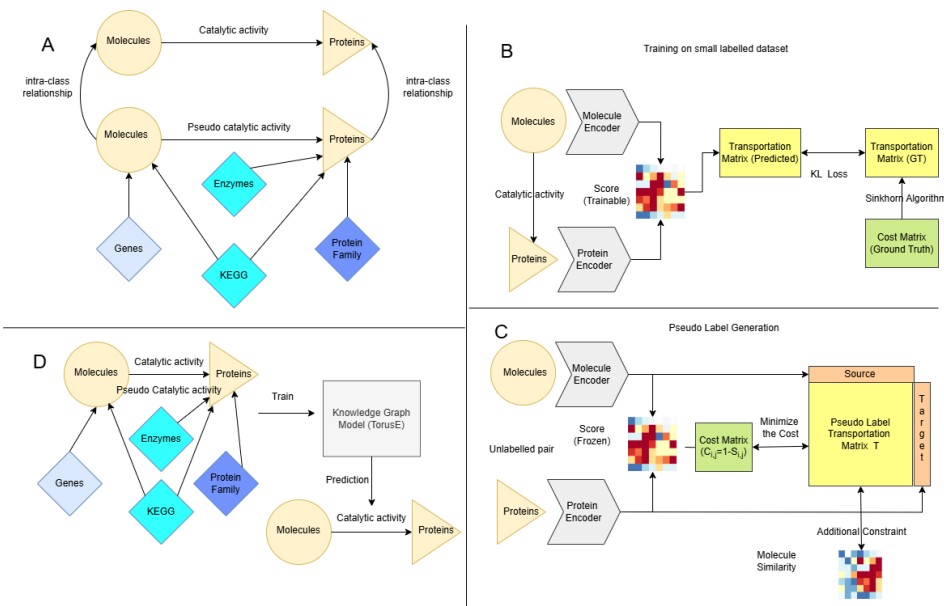

Figure 1: **Overview of KGOT. (A)** *Knowledge integration* **(B)** *Supervised score learning* **(C)** *Pseudo-label generation* **(D)** *KG augmentation & link prediction*

framework in which (1) an inverse optimal-transport (IOT) objective supervises mutual molecule–protein retrieval (Sec. 2.3), (2) the resulting OT transport plan is *written back* into the knowledge graph as a new `pseudo_interaction` relation, and (3) KG embeddings and retrieval models are trained jointly on the augmented graph (Sec. 2.4). This closes the loop between score learning, pseudo-label generation, and knowledge-graph training, enabling label-efficient MPI prediction with multimodal biological context.

Despite these advances, retrieving molecule-protein interactions remains a formidable challenge. A core issue is the scarcity of large-scale labeled datasets for MPIs, due to the experimental complexity and cost of validating interactions. Each new molecule-protein interaction must typically be confirmed via laborious assays, often with regulatory oversight in drug discovery, which severely limits data growth. High-throughput screening assays are expensive and slow, and even computational docking simulations (Di Nola et al., 1994) are constrained by accuracy and scale. Existing MPI datasets (Chandak et al., 2023; Gao et al., 2023) tend to be small, biased toward specific protein families, or inconsistent across annotations, making it difficult to train deep models that generalize across diverse interactions better than traditional techniques (such as docking and energy scoring (Wang et al., 2020)). If more labeled molecule-protein pairs were available spanning diverse chemistry and targets, deep learning models could learn richer interaction patterns and capture complex biophysical properties that traditional methods struggle with. Recent studies (Xia et al., 2024) underscore the need for novel frameworks that combine modern representation learning with strategies to cope with limited labels, in order to yield more robust and generalizable interaction predictions.

Another major challenge is the narrow reliance on *bi-modal* data, i.e., using only molecular and protein features while ignoring other relevant biological information. In real-world biology and drug discovery, molecular interactions are influenced by a broad spectrum of factors. For instance, genetic variations can alter protein function and a molecule's binding efficacy; biochemical pathways and networks can modulate the downstream effects of a drug and indicate which proteins are likely involved. These additional modalities provide essential context for understanding interactions beyond what comes with molecular structure or protein sequence alone.

Large-scale biological knowledge graphs (KGs) offer a promising way to integrate these heterogeneous modalities. Resources like PRIMEKG (Chandak et al., 2023) aggregate diverse biological entities and relations, but they contain relatively few direct molecule-protein interactions and are not tailored for MPI prediction. Our work aims to bridge this gap by systematically integrating multimodal biological data into the MPI prediction process. By using a biological KG as a structured

prior, we contextualize molecule-protein pairs with genetic, biochemical, and phenotypic insights, which can improve prediction robustness and generalization. In our framework, we extract relevant information from diverse sources into a unified graph and then refine the resulting representations through a pseudo-labeling mechanism based on optimal transport, aligning the multimodal knowledge with the MPI prediction task.

To address the data scarcity challenge, we constructed a large-scale *multimodal biological knowledge graph* by integrating six high-quality public datasets. The resulting KG contains over three million relations, encompassing molecules, proteins, genes, pathways, and other biomedical entities. By densely connecting molecule and protein nodes with diverse biological entities, this KG provides a rich relational context that compensates for the lack of direct molecule-protein labels. It also enables the model to capture indirect interaction paths, potentially improving generalization to new interactions.

Building on this knowledge graph, we propose a *pseudo-label generation framework* based on optimal transport to effectively leverage both labeled and unlabeled data. Instead of relying solely on the sparse interaction labels, we formulate pseudo-label assignment as a point-set matching problem: using OT, we align predicted interaction scores with the underlying biological structure to produce high-confidence pseudo-labels for unlabeled molecule-protein pairs. Concretely, our method first trains an interaction scoring function on top of molecule and protein encoders. We then infer scores for all unlabeled molecule-protein pairs to construct an initial dense score matrix. Next, we apply an OT-based algorithm to assign pseudo-label interaction probabilities by minimizing transport cost between molecule and protein distributions. The resulting pseudo-labeled interactions, combined with ground-truth labels, are used to augment training of the final model. This approach effectively transforms the MPI task into a semi-supervised learning problem, where the model benefits from an expanded training set of confident interaction examples.

In summary, our contributions are threefold:

- We construct a large-scale **multimodal knowledge graph** for MPI prediction by integrating diverse public datasets. This structured graph representation connects molecules and proteins through biological relationships, enabling the model to utilize multimodal context for interaction prediction.
- We propose an **optimal transport-based pseudo-labeling** strategy that treats label assignment as a point-set matching problem. This yields pseudo-labels aligned with the underlying data distribution and biological prior knowledge, improving the model's robustness and accuracy even with limited true labels.
- The **proposed framework achieves state-of-the-art performance** on multiple benchmark datasets, including virtual screening and MPI link prediction tasks. It outperforms prior approaches in terms of AUROC, early recognition metrics, and generalization to unseen interactions.

By bridging optimal transport with knowledge-driven representation learning, our approach provides a scalable and efficient solution for MPI prediction. Extensive experiments validate its effectiveness, particularly in leveraging multimodal biological signals to enhance interaction inference.

## 2 METHODOLOGY

### 2.1 OVERVIEW

We focus on the task of molecule–protein interaction prediction, more specifically, mutual retrieval of molecule-protein, which involves two complementary objectives: (1) Given a molecule $x$, retrieve the protein capable of best catalyzing it; (2) Given a protein $y$, retrieve the molecules that can best bind to it.

To address this task, we collect a biological knowledge graph dataset that integrates molecular, protein, and knowledge graph (KG)-based methodologies. We then propose a novel framework that leverages optimal transportation (OT) to generate pseudo-label of molecule-protein interactions. As shown in Figure 1, our approach is designed as follows:

(1) Collection of knowledge graph dataset. We collected over 1.35 million interactions between molecules and proteins from UniProt and CHEBI datasets, and more interactions related to genes,

genomes, protein families and enzyme commission numbers. Please check Appendix B for details of our dataset.

(2) Training on a small labeled dataset: We use the subset including only the molecule-protein pairs to train a scoring model that predicts the interaction strength for given pairs.

(3) Pseudo-label generation on a large unlabeled dataset: Using the trained model, we score molecule-protein pairs using all existing molecules and proteins in the dataset, these are mostly without pairwise labels and then we employ optimal-transport-based method to assign high-quality pseudo-labels.

(4) Knowledge graph augmentation: We extract the predicted molecule-protein pseudo-labels as supervision to the training on the full knowledge graph dataset for molecule-protein link predictions.

This framework enables the effective utilization of labeled and unlabeled data while leveraging relation information from KGs to enhance molecule-protein interaction predictions.

### 2.1.1 DESIGN CHOICES AND IMPLEMENTATION DETAILS.

**OT cost and marginals.** We define the OT cost as $C_{ij} = 1 - S_{ij}$, where $S_{ij}$ is the predicted interaction score between molecule $x_i$ and protein $y_j$ (see Section 2.3). Unless additional priors are available, we use uniform marginals $r_i = \frac{1}{M}$ and $c_j = \frac{1}{N}$, which encourages balanced coverage over molecules and proteins; we discuss alternative choices and robustness in Appendix E.

**Entropy regularization.** We solve the entropically regularized OT problem with Sinkhorn iterations. In our experiments we use $\epsilon = 0.01$ (see Appendix C) and additionally report a higher-entropy variant ($\epsilon = 0.1$) in the ablation study given in Appendix E.2.

**How the KG is encoded.** KGOT is agnostic to the underlying KG embedding model. We report results with a range of standard KGE architectures (PairRE, RotatE, MuRE, TorusE, ComplEx-FF; see Section 3.2), all trained on the same augmented KG with the additional `pseudo_interaction` relation (see Section 2.4).

**Leakage control.** For KG link prediction, we withhold a disjoint set of molecule–protein edges for evaluation and exclude them from both pseudo-label generation and KG training. For virtual screening, we additionally apply molecule- and protein-side filtering based on scaffold similarity and sequence identity (see Section 3.1, Appendix D).

### 2.2 PREPARATIONS FOR PSEUDO-LABEL GENERATION

**Pretrained backbone** To represent molecule and protein structural information effectively, we adopt pretrained encoders based on the Uni-Mol (Zhou et al., 2023) framework as the backbone for feature extraction. Uni-Mol is a molecule and protein pretraining model specially designed to process 3D conformations, and it has achieved strong performance on a variety of downstream tasks including molecular property prediction and binding pose prediction.

Two encoders produce embeddings $f(x)$ and $g(y)$ for molecule $x$ and protein $y$, respectively. These embeddings are normalized using the Euclidean norm to ensure consistency and compatibility for later tasks.

**Molecular similarity calculation** To measure the similarity between molecules, we utilize the embeddings extracted by the pretrained molecular encoder. Given two molecules $x_i$ and $x_j$, their respective embeddings $f(x_i)$ and $f(x_j)$ are first computed using the molecular encoder.

The similarity between molecules $x_i$ and $x_j$ is then quantified using the cosine similarity, which is defined as:

$$\text{Sim}(x_i, x_j) = \frac{\langle f(x_i), f(x_j) \rangle}{\|f(x_i)\|_2 \|f(x_j)\|_2},$$

where $\langle f(x_i), f(x_j) \rangle$ is the dot product of the two embeddings, and $\|f(x_i)\|_2$ and $\|f(x_j)\|_2$ are their respective Euclidean norms. This similarity measure serves as a foundation for incorporating molecular relationships into optimal transport-based pseudo-label generation.

## 2.3 Pseudo label generation with Optimal Transportation

To estimate interaction probabilities for molecule–protein pairs, we first train a scoring function $S(x, y)$ on the labeled subset. We extract features with pretrained encoders to obtain embeddings $f(x)$ and $g(y)$, and then compute scores for all molecule–protein pairs to form a score matrix $S \in \mathbb{R}^{M \times N}$ where $S_{ij} = S(x_i, y_j)$. Inspired by Shi et al. (2023), we model the global matching structure by learning an optimal transport plan $T \in \mathbb{R}^{M \times N}$ via:

$$\min_{T \geq 0} \sum_{i=1}^{M} \sum_{j=1}^{N} T_{i,j} C_{i,j}, \quad \text{s.t. } T\mathbf{1}_N = r, \ T^\top \mathbf{1}_M = c, \tag{1}$$

Here, the cost matrix $C \in \mathbb{R}_+^{M \times N}$, and $C_{i,j} = 1 - S_{i,j}$ shows the cost for pairing. Source and sink distributions are defined as: The source distribution $r \in \mathbb{R}^M$ satisfies $r_i = \frac{1}{M}$ and the sink distribution $c \in \mathbb{R}^N$ satisfies $c_j = \frac{1}{N}$. $\mathbf{1}_M, \mathbf{1}_N$ are all-one vectors with length of $M$ and $N$, respectively.

**Training strategy of score function on labeled dataset** To train the score function $S(x, y)$ effectively on a labeled dataset, we take an inverse optimal transportation perspective. The interaction score $S(x, y)$ between a molecule $x$ and a protein $y$ is modeled as a learned function of their embeddings:

$$S(x, y) = W(x \oplus y), \tag{2}$$

where $W$ is a trainable model, and $\oplus$ represents concatenation.

During training, we construct ground truth cost matrices $C_{\text{gt}}$ based on current batch of positive and negative molecule-protein pairs. For a given positive molecule-protein pair $(x_i, y_i)$, the ground truth cost $C_{\text{gt}}(i, j)$ is defined such that:

$$C_{\text{gt}}(i, j) = \begin{cases} 0, & \text{if } j = i \text{ (positive pair)} \\ 1, & \text{if } j \neq i \text{ (negative pair)} \end{cases}. \tag{3}$$

For a sampled batch of $N$ molecule-protein pairs, the theoretical optimal transport matrix $T_{\text{gt}}$ is computed using $C_{\text{gt}}$ as the cost matrix, following the Sinkhorn-Knopp algorithm to enforce marginal constraints:

$$T_{\text{gt}} = \arg\min_{T \geq 0} \sum_{i=1}^{N} \sum_{j=1}^{N} T_{i,j} C_{\text{gt}}(i, j), \quad \text{s.t. } T\mathbf{1}_N = r, \ T^\top \mathbf{1}_N = c, \tag{4}$$

where $r$ and $c$ are uniform source and sink distributions respectively.

Then we calculate the predicted transport matrix $T_{\text{pred}}$ based on the cost matrix derived from the predicted scores $C_{\text{pred}}(i, j) = 1 - S(x_i, y_j)$. The loss function is defined as the KL divergence between $T_{\text{pred}}$ and $T_{\text{gt}}$:

$$\mathcal{L}_{\text{score}} = \text{KL}(T_{\text{pred}} \| T_{\text{gt}}) = \sum_{i=1}^{N} \sum_{j=1}^{N} T_{\text{pred}}(i, j) \log \frac{T_{\text{pred}}(i, j)}{T_{\text{gt}}(i, j)}. \tag{5}$$

This formulation ensures that the learned score function $S(x, y)$ aligns the predicted transport matrix with the theoretical optimal transport matrix derived from ground truth labels. As described in Shi et al. (2023), contrastive learning with the InfoNCE loss can be viewed as a special form of this method. By minimizing $\mathcal{L}_{\text{score}}$, we get the optimal model $W$ that maximizes the scores of true molecule-protein pairs while minimizing those of false pairs in the batch.

**Pseudo label generation on large unlabeled dataset** Let us consider the scenario of the whole knowledge graph where we have $M$ molecules and $N$ proteins. The pairing degree between molecules and proteins is represented as a matrix $S \in \mathbb{R}_+^{M \times N}$, where each element $S_{i,j}$ reflects the score between molecule $i$ and protein $j$, calculated form the model we get in the former step. Our goal is to generate a pseudo-label matrix $T \in \mathbb{R}_+^{M \times N}$ that can be used for further training. We also treat this problem as an optimal transport problem.

We define the optimal transport problem as follows:

(1) The *cost matrix* $C \in \mathbb{R}_+^{M \times N}$ is defined as $C_{i,j} = 1 - S_{i,j}$, where a smaller cost corresponds to a higher pairing degree.

(2) The *source distribution* $r \in \mathbb{R}^M$ represents the initial distribution over the molecules, with each entry given by $r_i = \frac{1}{M}$.

(3) The *target distribution* $c \in \mathbb{R}^N$ represents the distribution over the proteins, with each entry given by $c_j = \frac{1}{N}$.

The optimal transport problem is to find a transportation matrix $T \in \mathbb{R}_+^{M \times N}$ that minimizes the total cost while satisfying the marginal constraints:

$$\min_{T \geq 0} \sum_{i=1}^{M} \sum_{j=1}^{N} T_{i,j} C_{i,j}, \text{subject to } T\mathbf{1}_N = r, \ T^\top \mathbf{1}_M = c, \tag{6}$$

where $\mathbf{1}_M$ and $\mathbf{1}_N$ are uniform distributions over lengths $M$ and $N$, respectively.

In our case, we have additional information about molecular similarity represented as a matrix $\text{Sim} \in \mathbb{R}_+^{M \times M}$, where $\text{Sim}_{i,k}$ quantifies the similarity between molecule $i$ and molecule $k$. To leverage this information, we introduce an additional constraint: the similarity of pseudo-labels between molecules $i$ and $k$, denoted by $\text{Sim}_{i,k}^T = \sum_{j=1}^{N} T_{i,j} T_{k,j}$, should be as close as possible to $\text{Sim}_{i,k}$.

The modified objective function becomes:

$$\min_{T \geq 0} \sum_{i=1}^{M} \sum_{j=1}^{N} T_{i,j} C_{i,j} + \lambda \sum_{i=1}^{M} \sum_{k=1}^{M} \left( \text{Sim}_{i,k} - \text{Sim}_{i,k}^T \right)^2, \tag{7}$$

where $\lambda > 0$ is a weighting factor balancing the cost and similarity terms, we take $\lambda = 0.1$

As shown in the algorithm, we employ the Sinkhorn-Knopp algorithm to solve the optimal transport problem efficiently and extend it to handle the similarity constraints.

The Sinkhorn-Knopp algorithm solves the regularized optimal transport problem by introducing an entropic regularization term:

$$\min_{T \geq 0} \sum_{i=1}^{M} \sum_{j=1}^{N} T_{i,j} C_{i,j} + \epsilon \sum_{i=1}^{M} \sum_{j=1}^{N} T_{i,j} \log T_{i,j}. \tag{8}$$

The solution can be computed iteratively: (1) Initialize $u = \mathbf{1}_M$, $v = \mathbf{1}_N$, and $K = \exp(-C/\epsilon)$; (2) Iterate until convergence: $u \leftarrow \frac{r}{Kv}, v \leftarrow \frac{c}{K^\top u}$; (3) Compute $T$ as: $T = \text{diag}(u) K \text{diag}(v)$. We take $\epsilon = 0.01$ for the experiments. Ablations can be found in Appendix E.

To incorporate similarity constraints, we modify $T$ using gradient-based optimization. The gradient of the similarity term with respect to $T$ is given by:

$$\nabla_{T_{i,j}} = 2\lambda \sum_{k=1}^{M} \left( \text{Sim}_{i,k} - \text{Sim}_{i,k}^T \right) T_{k,j}. \tag{9}$$

The algorithm alternates between updating $T$ using the Sinkhorn steps and refining $T$ with the similarity constraint gradient:

$$T \leftarrow T - \eta \nabla_T, \tag{10}$$

where $\eta > 0$ is the learning rate, and $T$ is projected back to the feasible set if needed.

**Pseudo-label extraction from the OT plan.** After solving the regularized OT problem, we obtain a transport plan $T \in \mathbb{R}_+^{M \times N}$ that assigns a soft matching mass $T_{ij}$ to each molecule–protein pair $(x_i, y_j)$. We treat entries with large transport mass as high-confidence pseudo-positive interactions:

$$\mathcal{P}_\delta = \{(x_i, y_j) \mid T_{ij} \geq \delta\}, \tag{11}$$

we use $\delta = 0.5$ in all main experiments unless stated otherwise. In practice, $\delta$ controls a precision–coverage trade-off: a larger $\delta$ yields fewer but cleaner pseudo-labels, while a smaller $\delta$ increases coverage but may introduce noise.

---

**Algorithm 1:** Training strategy on large unlabeled dataset

---

**Input:** Pairing score matrix $S \in \mathbb{R}_+^{M \times N}$, molecular similarity matrix $\mathrm{Sim} \in \mathbb{R}_+^{M \times M}$, source distribution $r \in \mathbb{R}^M$, target distribution $c \in \mathbb{R}^N$, entropic regularization parameter $\epsilon$, similarity weight $\lambda$, learning rate $\eta$, maximum iterations max_iter.

**Output:** Optimal transport matrix $T \in \mathbb{R}_+^{M \times N}$.

**Initialization:**

Define cost matrix $C \in \mathbb{R}_+^{M \times N}$ as $C_{i,j} = 1 - S_{i,j}$.

Set $K = \exp(-C/\epsilon)$, $u = \mathbf{1}_M$, $v = \mathbf{1}_N$, and $T = 0$.

**for** $t = 1$ *to max_iter* **do**

    **Step 1: Sinkhorn-Knopp Iteration.**

    **while** *not converged* **do**

        Update $u \leftarrow \frac{r}{Kv}$.

        Update $v \leftarrow \frac{c}{K^\top u}$.

    Compute $T = \mathrm{diag}(u) K \mathrm{diag}(v)$.

    **Step 2: Similarity Constraint Adjustment.**

    Compute $\mathrm{Sim}_{i,k}^T = \sum_{j=1}^N T_{i,j} T_{k,j}$ for all $i, k$.

    Compute gradient $\nabla_{T_{i,j}} = 2\lambda \sum_{k=1}^M \left( \mathrm{Sim}_{i,k} - \mathrm{Sim}_{i,k}^T \right) T_{k,j}$.

    Update $T \leftarrow T - \eta \nabla_T$.

    **Step 3: Projection onto Feasible Set.**

    Ensure $T \geq 0$, and normalize $T$ such that $T\mathbf{1}_N = r$ and $T^\top \mathbf{1}_M = c$.

**return** $T$.

---

Table 1: Results on the DUD-E virtual screening benchmark (zero-shot setting). Higher is better for all metrics. Our OT + KG framework outperforms both traditional docking methods and modern learning-based approaches across all evaluation metrics.

| Model | AUROC (%) | BEDROC (%) | EF@0.5% | EF@1% | EF@2% |
|---|---|---|---|---|---|
| Glide-SP (Halgren et al., 2004) | 76.70 | 40.70 | 19.39 | 16.18 | 7.23 |
| Vina (Trott and Olson, 2010) | 71.60 | – | 9.13 | 7.32 | 4.44 |
| NN-score (Durrant and McCammon, 2011) | 68.30 | 12.20 | 4.16 | 4.02 | 3.12 |
| RFscore (Ballester and Mitchell, 2010) | 65.21 | 12.41 | 4.90 | 4.52 | 2.98 |
| Pafnucy (Stepniewska-Dziubinska et al., 2018) | 63.11 | 16.50 | 4.24 | 3.86 | 3.76 |
| OnionNet (Zheng et al., 2019) | 59.71 | 8.62 | 2.84 | 2.84 | 2.20 |
| Planet (Zhang et al., 2023) | 71.60 | – | 10.23 | 8.83 | 5.40 |
| DrugCLIP (Jia et al., 2026) | 80.93 | 50.52 | 38.07 | 31.89 | 10.66 |
| **KGOT** | **83.45 ± 0.42** | **51.20 ± 0.35** | **39.10 ± 0.50** | **33.00 ± 0.47** | **11.20 ± 0.30** |

## 2.4 UNIFIED FRAMEWORK FOR LINK PREDICTION

To enhance the protein-molecule link prediction task, we propose a unified framework that integrates the pseudo-label matrix $T$, generated by the optimal transport-based training strategy, with the structured knowledge from the knowledge graph (KG).

The relations in KG dataset include all observed interactions in the KG as well as a new relation type, `pseudo_interaction`, which encodes the pseudo-label scores $T$.

**Multi-objective learning framework** Our model is optimized with a multi-objective loss that jointly leverages the knowledge graph structure and pseudo-label supervision. Specifically, we combine a graph embedding loss over KG triples with a pseudo-label alignment term that encourages predicted interaction scores to match the generated pseudo-label matrix. This joint formulation allows the model to balance structural knowledge with data-driven signals. Full mathematical definitions and implementation details are provided in Appendix F.

## 3 EXPERIMENTS AND RESULTS

Our evaluation spans two settings that highlight different aspects of the proposed framework: (1) virtual screening benchmarks, which test the model's ability to retrieve active molecules for given

Table 2: Results on the LIT-PCBA benchmark (zero-shot setting). Our method achieves the best performance across all metrics, illustrating its robustness on this more challenging dataset.

| Model | AUROC (%) | BEDROC (%) | EF@0.5% | EF@1% | EF@5% |
|---|---|---|---|---|---|
| Surflex (Jain, 2003) | 51.47 | – | – | 2.50 | – |
| Glide-SP (Halgren et al., 2004) | 53.15 | 4.00 | 3.17 | 3.41 | 2.01 |
| Planet (Zhang et al., 2023) | 57.31 | – | 4.64 | 3.87 | 2.43 |
| Gnina (McNutt et al., 2021) | 60.93 | 5.40 | – | 4.63 | – |
| DeepDTA (Öztürk et al., 2018) | 56.27 | 2.53 | – | 1.47 | – |
| BigBind (Brocidiacono et al., 2023) | 60.80 | – | – | 3.82 | – |
| DrugCLIP (Jia et al., 2026) | 57.17 | 6.23 | 8.56 | 5.51 | 2.27 |
| **KGOT** | **62.45 ± 0.38** | **6.52 ± 0.22** | **9.12 ± 0.40** | **5.90 ± 0.28** | **2.50 ± 0.15** |

protein targets in a zero-shot manner, and (2) knowledge graph link prediction, which tests the model's ability to identify held-out molecule–protein links using the integrated KG.

### 3.1 EVALUATION ON VIRTUAL SCREENING TASKS

Virtual screening benchmarks evaluate how well models can rank candidate molecules for a particular protein target, based on predicted binding likelihood. We consider two widely-used datasets: **DUD-E** and **LIT-PCBA**. We follow the evaluation protocol of recent works like DrugCLIP (Gao et al., 2023) to ensure fair comparison.

**DUD-E benchmark.** The DUD-E dataset (Mysinger et al., 2012) contains 102 protein targets, each with a set of known active molecules (binders) and a large set of decoys (inactive molecules). In total, DUD-E includes 22,886 active molecule–target pairs. We frame each target's screening as a ranking task: the model must assign higher scores to active molecules than to decoys. We evaluate performance using three metrics commonly used in virtual screening: (1) **AUROC** (area under the ROC curve), a threshold-independent measure of ranking quality; (2) **BEDROC** with $\alpha = 20$, which emphasizes early recognition of actives (important in screening scenarios); (3) **Enrichment Factor (EF)** at certain top fractions (e.g., EF@0.5%, 1%, 2%), which measures how many actives are found among the top-ranked subset compared to random expectation.

**Leakage control.** To preclude train–test leakage on the *molecule side*, we remove from the training pool any ligand whose Tanimoto similarity to *any* DUD-E test active is above a threshold of $\geq 0.60$; we additionally report a stricter *Murcko-scaffold–out* variant in which all training ligands sharing the Bemis–Murcko scaffold with any test active are excluded (see Appendix D). On the *protein side*, we exclude from training any protein whose sequence identity (computed by MMseqs2) to *any* DUD-E test target exceeds 60%; we also provide a *family–out* control by removing all training proteins mapped (via HMMER to Pfam-A) to the same families as DUD-E test targets. We use a single model (no ensembling or post-hoc re-ranking) for all runs.

Our model produces an interaction score for each molecule–target pair, which we use to rank molecules for each target. Table 1 summarizes the results on DUD-E in the zero-shot setting. We compare to classical docking methods such as *Glide-SP* (Halgren et al., 2004) and *AutoDock Vina* (Trott and Olson, 2010), and learned baselines including similarity-based models such as *NN-score* (Durrant and McCammon, 2011) and *RFscore* (Ballester and Mitchell, 2010), structure-based deep models such as *Pafnucy* (Stepniewska-Dziubinska et al., 2018) and *OnionNet* (Zheng et al., 2019), and recent cross-modal representation models such as *Planet* (Zhang et al., 2023) and *DrugCLIP* (Jia et al., 2026). Our method achieves the highest scores on all metrics. Notably, it outperforms the previous best model by a significant margin in early recognition metrics (e.g., BEDROC and EF), indicating that the OT-guided pseudo-labeling helps identify actives at the top of the ranking more effectively. In terms of AUROC, we obtain about 83.5%, which is an improvement of ∼2.5% over DrugCLIP (80.9%) and substantially higher than traditional docking (Glide SP: 76.7%). These results demonstrate the benefit of augmenting the limited training interactions with pseudo-labeled examples and KG-derived information. Here we also report mean ± standard deviation over multiple runs with 3 different random seeds

**LIT-PCBA benchmark.** LIT-PCBA is another benchmark from the Therapeutic Data Commons, designed to be more challenging and address some biases in DUD-E. It comprises 15 protein targets

with 7,844 experimentally confirmed actives and 407,381 inactives. The class imbalance is even more extreme here, making early retrieval metrics critical. We evaluate our model on LIT-PCBA under the same zero-shot setting.

Table 2 shows the results. We compare to several docking and deep learning baselines reported for this benchmark, including Surflex (Jain, 2003), Glide-SP, Gnina (McNutt et al., 2021), DeepDTA (Öztürk et al., 2018), BigBind (Brocidiacono et al., 2023), and DrugCLIP. Our framework again achieves the top performance on all metrics. In particular, we see improvements in AUROC (62.45% vs. 60.93% for the best baseline, Gnina) and in early enrichment (EF@0.5% of 9.12 vs. 8.56 for DrugCLIP). Although the absolute values are lower than DUD-E (due to LIT-PCBA's difficulty), the consistent gains indicate that our pseudo-label + KG approach generalizes well.

## 3.2 MOLECULE-PROTEIN LINK PREDICTION TASK

We next evaluate our unified framework on the task of predicting molecule–protein links on knowledge graph. For this, we created a held-out set of known molecule–protein interactions from our collected KG data. Specifically, we withhold 60,000 molecule–protein pairs (edges) from the KG to serve as test examples, these pairs were not included in the training pseudo-label generation or KG training. Each test pair is a true interaction. The model must predict these links purely from the remaining training data in the KG and the pseudo-label enriched representations.

We cast this as a link prediction problem: given a molecule node, rank candidate protein nodes by the predicted likelihood of interaction (head entity prediction in KG terms). We evaluate using standard information retrieval metrics Hit@K: Hits@1, Hits@3, and Hits@5, which measure the proportion of test queries for which the correct partner appears in the top 1, top 3, or top 5 predictions, respectively. These metrics reflect the model's ability to place the true interaction at or near the top of the candidate list.

We compare our full model (which uses pseudo-labels and KG, as described) against ablated versions to quantify the effect of pseudo-labeling. In Table 3, we report results for several knowledge graph embedding models with and without our pseudo-label augmentation: PairRE, RotatE, MuRE, TorusE, and ComplEx-FF are five different KG embedding architectures. For each, we train one model using only the real KG edges (baseline) and another including the `pseudo_interaction` edges and the alignment loss. We observe that incorporating pseudo-labels leads to consistent improvements across all model types. The absolute gains vary by model, but the trend is clear: the additional pseudo-labeled interactions help the KG embedding models better discriminate true links. Among the embedding methods, we found TorusE performed strongly, and with pseudo-labels it achieved the highest Hits@5 (74.9%). ComplEx-FF benefited remarkably from pseudo-labels (Hits@1 rising from 30.8% to 43.6%). These results highlight that our pseudo-labeling approach is model-agnostic and can enhance a range of link prediction algorithms by providing extra supervision.

Table 3: Knowledge graph link prediction performance (Hits@K) with and without KGOT generated pseudo-label augmentation.

| Method | Hits@1 | Hits@3 | Hits@5 |
|---|---|---|---|
| PairRE | 10.0% | 17.0% | 21.4% |
| PairRE + KGOT | **10.9%** | **20.5%** | **26.4%** |
| RotatE | 48.5% | 61.6% | 66.6% |
| RotatE + KGOT | **52.0%** | **63.9%** | **68.0%** |
| MuRE | **15.9%** | 24.7% | 31.0% |
| MuRE + KGOT | 13.7% | **25.9%** | **31.2%** |
| TorusE | 49.4% | 64.2% | 70.0% |
| TorusE + KGOT | **53.4%** | **65.2%** | **74.9%** |
| ComplEx-FF | 30.8% | 40.2% | 44.4% |
| ComplEx-FF + KGOT | **43.6%** | **54.3%** | **58.6%** |

The consistent improvements demonstrate that our pseudo-label generation successfully injects useful information into the KG. By effectively "filling in" likely interactions that were not explicitly in the KG, the model can learn from a more complete interaction network. This leads to better retrieval of held-out interactions, validating the core idea of our approach: leveraging unlabeled pairs through OT-based pseudo-labeling boosts prediction performance.

### 3.3 ABLATION STUDY.

We conduct ablation experiments to assess the contributions of our design choices, including (i) the use of OT loss versus standard InfoNCE contrastive loss, (ii) different pseudo-labeling strategies, and (iii) the integration of multiple biological knowledge sources. Across all settings, our proposed OT-based formulations and multimodal knowledge graph integration consistently yield higher link prediction accuracy. For example, OT loss improves over InfoNCE, OT+similarity pseudo-labeling achieves the best Hits@5, and adding GO, protein family, and pathway relations each provide incremental gains in Hits@1. Due to space limit, full details and complete results are reported in Appendix E.

## 4 CONCLUSION

In this work, we provide a unified biomedical knowledge graph model to tackle the challenge of molecular-protein interaction retrieval by integrating multi-modal biological data to improve molecule–protein interaction prediction and proposing a novel pseudo-label generation framework based on optimal transport (OT) to mitigate the scarcity of large-scale labeled datasets. Our approach integrates multiple biological datasets into a unified knowledge graph, spanning drugs, proteins, genes, and biological processes, aligning predicted interaction distributions with the underlying graph structure, significantly improving retrieval performance across multiple benchmark datasets.

Extensive experiments validate the superiority of our approach, demonstrating consistent performance gains across various molecular-protein interaction prediction tasks.

Overall, our work presents a scalable and efficient framework for molecular-protein interaction retrieval, bridging the gap between structured biological knowledge and deep learning-based representation learning. We hope that our work can provide a paradigm for other data-lacking tasks in the field of computational biology, and ultimately point the way to using machine learning methods to build unified foundation models in the biological field.

## 5 RECOMMENDED ATTACHMENTS

### 5.1 ETHICS STATEMENT

This work uses publicly available biomedical resources to construct a multimodal knowledge graph and evaluate molecule–protein interaction prediction on established benchmarks (DUD-E, LIT-PCBA). No human subjects, PHI/PII, or clinical interventions are involved. We follow the licenses of all data providers.

### 5.2 REPRODUCIBILITY STATEMENT

We provide end-to-end details to enable reproduction: data sources, entity/edge schemas, and counts for the integrated KG (Appendix B); leakage-control splits and zero-shot protocols for DUD-E and LIT-PCBA (Section 3.1, Appendix D); the OT-based pseudo-label generation objective and algorithm (Section 2.3, Alg. 1), along with hyper-parameters and hardware specs (Appendix C). An anonymous repository with code and data is referenced in the appendix.

### 5.3 USAGE OF LLM

LLMs were not used to generate data, labels, or experimental results. The core methodology relies on pretrained molecular/protein encoders and an optimal-transport pseudo-labeling framework integrated with a biomedical KG. LLMs were used only as general-purpose assist tools for light copy-editing and minor code polishing; all technical ideas, algorithms, experiments, and analyses were conceived and implemented by the authors, who take full responsibility for accuracy and integrity.

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

# A  RELATED LITERATURE

## A.1  OPTIMAL TRANSPORT IN REPRESENTATION LEARNING

Optimal Transport (OT) is a mathematical framework originally developed to solve resource allocation problems by minimizing the cost of transporting mass from one distribution to another. Recently, it has been increasingly applied in machine learning and representation learning tasks due to its ability to compare distributions in a geometrically meaningful way. Unlike traditional distance metrics such as Euclidean or cosine distances, OT considers the geometry of the distributions, enabling it to capture fine-grained relationships between data points.

In the context of multimodal representation learning, OT has been used to align embeddings from different modalities by computing an optimal coupling between them. This approach ensures that structurally similar elements across modalities are closely matched, thus enhancing the quality of learned representations. Applications of OT have been particularly impactful in cross-modal tasks, such as image-text retrieval, molecular-protein interaction modeling, and domain adaptation. Notable works include the Sinkhorn-Knopp algorithmSinkhorn and Knopp (1967), which makes OT computationally efficient for large-scale datasets by introducing entropy regularization to the transport problem. Shi et al. (2023) aim to understand and generalize contrastive learning as a form of inverse optimal transport. The paper also shows that InfoNCE contrastive loss is a specific case of the proposed IOT loss. Other notable works in applying Optimal Transportation in feature learning include Fan et al. (2024), Lee et al. (2022),Qin et al. (2024) and Gossi et al. (2023)

## A.2  KNOWLEDGE GRAPHS IN MULTI-MODALITY MOLECULAR AND PROTEIN TASKS

Knowledge graphs (KGs) have emerged as powerful tools for integrating and modeling complex, heterogeneous data in multi-modality tasks, including molecular and protein-related studies. A KG represents entities (e.g., molecules, proteins, and biological pathways) as nodes and their relationships (e.g., binding, inhibition, or interaction) as edges, enabling the incorporation of prior biological knowledge into machine learning models.

In molecular and protein studies, KGs such as DrugBankKnox et al. (2024), ChEMBLGaulton et al. (2012), and STRINGMering et al. (2003) have been widely used to capture molecular-protein interactions and other biological relationships. By encoding domain-specific knowledge into graph structures, KGs enhance downstream tasks like molecular property predictionFang et al. (2023), drug-drug interaction predictionLin et al. (2020), and disease-gene association studiesVilela et al. (2023).

Recent advancements have also explored cross-modality tasks involving molecules and proteins by leveraging KGs as a common representation space. For example, BioBridgeWang et al. (2024) aligns molecule and protein embeddings through a knowledge graph and evaluates cross-domain retrieval tasks. Song et al. (2026) is another application of knowledge graphs in biological modeling. Similarly, graph neural networks (GNNs) have been applied to propagate information across graph nodes, enabling the extraction of contextual embeddings that incorporate relational knowledge. Despite these successes, integrating KGs with multi-modality data remains challenging due to the heterogeneous nature of molecular and protein features, as well as the sparsity of some entity relationships.

**Positioning with KG-based MPI frameworks.**  Several recent KG-based approaches are closely related to our setting but differ in their supervision and how they exploit KG structure. (Ma et al., 2022) propose **KG-MTL**, which jointly leverages a biomedical KG (for drug entities) and molecular graphs (for compound structures) under a multi-task learning objective for DTI/CPI prediction. (Ma et al., 2024) propose **BioKDN**, a robustification module that *denoises* local KG neighborhoods and smooths relation semantics to mitigate noisy edges when predicting molecular interactions. In contrast, **KGOT** is designed for *unlabeled or weakly-labeled MPI retrieval*: we (i) construct a *unified, multimodal* KG spanning compounds, proteins, pathways, GO terms, and protein families, (ii) generate *global* pseudo-labels via entropy-regularized optimal transport that enforces dataset-level matching constraints, and (iii) *write back* high-confidence pseudo interactions into the KG to enable downstream KG completion.

| Method | Unified multimodal KG | KG denoising | Multi-task learning | Global OT pseudo-labeling |
|---|---|---|---|---|
| KG-MTL Ma et al. (2022) | × | × | ✓ | × |
| BioKDN Ma et al. (2024) | × | ✓ | × | × |
| KGOT (ours) | ✓ | × | × | ✓ |

Table 4: Qualitative comparison between representative KG-based molecular interaction prediction methods and KGOT. KGOT targets weakly-/unlabeled MPI retrieval via global OT pseudo-labeling and KG write-back, rather than multi-task supervised learning or KG denoising.

## B  DETAILS OF DATASET CONSTRUCTION

To develop a comprehensive knowledge graph to study molecule and protein interactions, we considered 6 primary resources of biological and clinical information. The data resources provide widespread coverage of biomedical entities, including proteins, genes, drugs, molecules, biological processes, cellular components and protein families. These were high-quality datasets, either expertly curated annotations such as KEGG, widely-used standardized ontologies such as the Gene Ontology, or direct readouts of existing large scale unimodality dataset, such as CHEBI and UniProt.

(1) 1.35 million potential catalytic activity relationships between over 30,000 molecules from CHEBIDegtyarenko et al. (2007) and 500,000 proteins from UniProtUniProt Consortium (2018).

(2) Genetic and genomic information from KEGGKanehisa and Goto (2000), capturing how molecular interactions are influenced by gene regulation and metabolic pathways.

(3) Functional annotations from Gene Ontology Ashburner et al. (2000), enriching molecular representations with biological process and cellular component insights.

(4) Protein family classifications from PFaMFinn et al. (2014), improving interaction predictions by incorporating shared functional domains.

(5) Enzyme Commission numbers from ENZYMEBairoch (2000), allowing for enzyme-substrate relationship modeling within our graph.

| Head Entities | | Tail Entities | |
|---|---|---|---|
| Type | Quantity | Type | Quantity |
| UNIPROT | 5,956,325 | GO | 3,191,321 |
| CHEBI | 336,374 | CHEBI | 1,678,407 |
| KEGG | 92,184 | PFAM | 792,235 |
| GO | 89,235 | KEGG_KO | 407,307 |
| EC | 8,459 | EC | 304,428 |
| KMODUL | 1,275 | KPATHWAY | 89,989 |
| | | KCOMPOUND | 16,695 |
| | | KMODULE | 3,470 |

Table 5: Entities Distribution, where KMODUL represents KEGG_MODUL, KPATHWAY represents KEGG_PATHWAY, KCOMPOUND represents KEGG_COMPOUND, and KMODULE represnts KEGG_MODULE.

Our knowledge graph dataset is a multimodal knowledge graph with 8 types of nodes, 29 types of directed edges, 6,483,852 relationships between entities.

## C  IMPLEMENTATION DETAILS

**Backbone architecture.**  We use Uni-Mol Zhou et al. (2023) for both molecular and protein encoders. Hidden dimension is set to 512 for both. The scoring MLP has two layers of size [512, 256, 1], with ReLU activations.

**Labeled dataset training.**    Training on the labeled set uses a batch size of 128, learning rate 1e-4, and Adam optimizer with weight decay 0.01. The score function $S(x, y)$ is trained for 50 epochs with early stopping.

**Pseudo-label generation.**    For full-batch OT, we construct a score matrix for 10k molecules × 5k proteins. Sinkhorn $\epsilon = 0.01$, similarity weight $\lambda = 0.1$, learning rate $\eta = 1.0$, and 50 iterations. Top-$k$ baseline selects the top 5 pseudo-labels per protein.

**Knowledge graph training.**    KG embeddings are trained with embedding size 256 and margin 6.0. We use 1:1:1 sampling for real:negative:pseudo triples and a batch size of 1024. Pseudo-interactions are weighted via the loss term $\mathcal{L}_{\text{pseudo}}$.

**Hardware.**    All experiments are run on 4 × NVIDIA A6000 48GB GPUs with 256GB RAM. Total training time for each variant is under 12 hours.

## D    LEAKAGE-CONTROL EXPERIMENTS

**Protocol.**    To further mitigate train–test leakage beyond the default setup in the main text, we evaluate our model under a *strict* protocol that combines molecule- and protein-side filtering: (i) on the **molecule side**, we adopt a *Murcko-scaffold–out* setting where all training ligands sharing the Bemis–Murcko scaffold with any test active are removed; (ii) on the **protein side**, we adopt a *family–out* setting by removing from the training pool all proteins that map (via HMMER to Pfam-A) to any family present among test targets. All other training, inference, and evaluation details (single model, zero-shot evaluation, fixed seeds) follow the main-text configuration.

**Results on DUD-E (strict leakage control).**    Table 6 reports performance under the combined *Murcko-scaffold–out* (ligands) + *Pfam family–out* (proteins) protocol. As expected, absolute numbers are lower than in Table 1, yet early recognition remains strong under strict filtering.

Table 6: DUD-E under strict leakage control.

| Model | AUROC (%) | BEDROC (%) | EF@0.5% | EF@1% | EF@2% |
|---|---|---|---|---|---|
| KGOT (strict) | 81.78 | 51.04 | 38.91 | 32.47 | 10.35 |

**Results on LIT-PCBA.**    Table 7 summarizes performance on LIT-PCBA under the same strict protocol. Given the greater class imbalance and challenge level of LIT-PCBA, we continue to emphasize early enrichment metrics.

Table 7: LIT-PCBA under strict leakage control

| Model | AUROC (%) | BEDROC (%) | EF@0.5% | EF@1% | EF@5% |
|---|---|---|---|---|---|
| KGOT (strict) | 61.22 | 5.96 | 8.92 | 5.34 | 2.27 |

## E    ABLATION EXPERIMENTS

### E.1    OT LOSS VS. CONTRASTIVE LOSS (INFONCE) IN SUPERVISED TRAINING

We first compare the proposed optimal transport loss to a standard contrastive loss (InfoNCE) for supervising the scoring model. This experiment evaluates whether our OT-based loss offers an advantage over a more conventional pairwise contrastive approach.

Experimental Setup: We train the scoring model (MuRE-based encoder) on known entity pairs using either (i) our OT loss, or (ii) an InfoNCE loss. For InfoNCE, each positive pair is contrasted against

multiple randomly sampled negative pairs (with temperature tuned to 0.1). All other training settings (learning rate, epochs, etc.) are kept identical. We evaluate the models on the link prediction task.

Following Table shows the performance comparison. The model trained with the OT loss achieves slightly higher accuracy than with InfoNCE. For instance, OT loss yields an MRR of 0.256 vs. 0.243 with InfoNCE, and Hits@10 of 45.1 vs. 42.7.

| Training Loss | MRR | Hits@10 |
|---|---|---|
| OT Loss (ours) | 0.256 | 45.1% |
| InfoNCE Loss | 0.243 | 42.7% |

Table 8: Performance of the scoring model with OT-based loss vs. contrastive InfoNCE loss. OT loss yields a modest but consistent improvement in link prediction metrics.

The OT-trained model outperforms the InfoNCE variant, indicating that the OT formulation provides a beneficial supervisory signal. We attribute this gain to the OT loss's ability to consider the full distribution of true associations for each entity, rather than only one positive against negatives at a time. In knowledge graph link prediction, an entity can have multiple correct targets; the OT loss naturally accommodates multiple positive matches by optimizing a transport plan over them.

### E.2 Pseudo-labeling strategies

We compare no pseudo labels, random selection, top-$k$ per protein, OT without similarity, high-entropy OT, and our full OT+similarity.

OT yields balanced pseudo labels with broader protein/molecule coverage than top-$k$, while the similarity prior filters implausible assignments. Excessive entropy ($\epsilon$ large) makes labels diffuse and less informative.

### E.3 Components Ablation

To further analyze the contributions of different components, we perform an ablation on the knowledge graph variant of our model. Table 10 shows an ablation using the TorusE embedding method on the link prediction task, where we incrementally add sources of information. Starting with only molecule-protein edges (using the small labeled set, akin to a baseline without any additional knowledge), we then add gene ontology (GO) relations, protein family relations, and metabolic process (pathway) relations from the knowledge graph. Each addition yields an improvement in Hits@1. Specifically, incorporating GO relations (which provide gene-function context) boosts Hits@1 from 36.3% to 43.7%; adding protein family info further raises it to 47.4%; and including metabolic pathways brings it to 53.4%. This ablation highlights that each modality of biological knowledge contributes to better performance. It underscores the importance of multimodal data integration: the model with full knowledge (last row) performs significantly better than using only the labeled molecule-protein pairs.

## F Multi-Objective Learning Framework

The framework is trained using a multi-objective loss function that integrates information from both the pseudo label matrix and the KG structure. The loss components are as follows:

**Graph embedding loss** A knowledge graph embedding model is trained to learn representations of entities and relations in the KG. The graph embedding loss is defined as:

$$\mathcal{L}_{\text{KG}} = \sum_{(h,r,t) \in \mathcal{D}_{\text{KG}}} \log \sigma(f(h,r,t)) \quad + \sum_{(h',r,t') \notin \mathcal{D}_{\text{KG}}} \log \sigma(-f(h',r,t')), \tag{12}$$

where $f(h,r,t)$ is the scoring function for a triple $(h,r,t)$, $\sigma$ is the sigmoid function, and $\mathcal{D}_{\text{KG}}$ and $\mathcal{D}'_{\text{KG}}$ are the sets of positive and negative samples, respectively.

Table 9: Ablation on pseudo-label generation (illustrative Hits@5 on link prediction).

| Strategy | Hits@5 (%) |
|---|---|
| No pseudo-label augmentation | 68.5 |
| Random pseudo-labels (matched count) | 66.0 |
| Top-$k$ per protein ($k = 5$) | 72.0 |
| OT without similarity ($\lambda = 0$) | 73.8 |
| OT with high entropy ($\epsilon = 0.1$) | 73.0 |
| **OT + similarity ($\lambda = 0.1,\ \epsilon = 0.01$)** | **74.9** |

Table 10: Ablation study on effect of integrating different knowledge graph relation types (using TorusE).

| Model Configuration | Hits@1 (Link Prediction) |
|---|---|
| TorusE (molecules & proteins only) | 36.3% |
| + Gene Ontology relations | 43.7% |
| + Protein family relations (GO + PF) | 47.4% |
| + Metabolic process relations (GO + PF + KEGG + EC) | **53.4%** |

**Pseudo label alignment loss**   The pseudo labels $T$ represent predicted scores for protein-molecule interactions. We incorporate these into the model by defining a loss term that aligns the KG-predicted scores with $T$:

$$\mathcal{L}_{\text{pseudo}} = \sum_{i=1}^{M} \sum_{j=1}^{N} \left( f(e_i, r_{\text{pseudo}}, e_j) - T_{i,j} \right)^2, \tag{13}$$

where $e_i$ and $e_j$ are the embeddings of molecule $i$ and protein $j$, respectively, and $r_{\text{pseudo}}$ is the learned embedding for the pseudo_interaction relation.

The total loss function for training the model is a weighted sum of the above components:

$$\mathcal{L}_{\text{total}} = \mathcal{L}_{\text{KG}} + \alpha \mathcal{L}_{\text{pseudo}}, \tag{14}$$

where $\alpha$ is the hyperparameter that balance the contributions of the pseudo label and similarity terms, we take $\alpha = 0.1$ in the experiments.

