# OpenReview forum: "KGOT: Unified Knowledge Graph and Optimal Transport Pseudo-Labeling for Molecule-Protein Interaction Prediction"
_ICLR.cc/2026/Conference — ICLR 2026 Poster_

### Official Review · Reviewer_Tsmw · 2025-10-27

**Soundness:** 3
**Presentation:** 3
**Contribution:** 2
**Rating:** 4
**Confidence:** 5

**Summary:**

In the paper, the authors studied methods for predicting ligand-target binding affinity as link prediction problems. In order to resolve the issue of lacking large-scale training data. The authors proposed methods for augmentation of labelled data with useful information from multi-modal knowledge graphs.

The proposed method following 4 different steps:

+ use a small labelled datasets to train a predictive model that predicts the probability of binding between any pair of molecule and protein.

+ use the given model to create a pseudo-label data on a larger unlabelled graph. The pseudo labels are enforced to give overall consistency prediction on the entire unlabelled graph by Optimal Transport and the regularization of the similarity between nodes on the latent spaces.

+ the pseudo-labelled graph together augmented with the edges of the large multimodal knowledge graph is then used to train a final link prediction model

In order to evaluate the proposed approach the author compared there KGOT method with baseline methods that do not use external information on the DUDe and  LIT-PCBA with leakage removal from the training graphs. Their proposed approach yields better results then the baseline methods.

In another experiments for link prediction, KGOT was used in addition to other graph embedding methods. the experimental results show that the data augmentation procedure in KGOT via OT helps simple graph embedding methods improve linking prediction accuracy.

**Strengths:**

An interesting idea was proposed regarding using OT to control the consistency of pseudo-labels overall.

The paper was well written and easy to follow.

**Weaknesses:**

The idea of using multimodal knowledge graphs to augment training data for ligand-target affinity prediction is not a new idea. The given idea has been proposed and studied in the following works:

+ N. Zhang, Z. Bi, X. Liang, S. Cheng, H. Hong, S. Deng, Q. Zhang, J. Lian, and H. Chen. Ontoprotein: Protein
pretraining with gene ontology embedding. In International Conference on Learning Representations, 2022.

+ H.-Y. Zhou, Y. Fu, Z. Zhang, B. Cheng, and Y. Yu. Protein representation learning via knowledge enhanced
primary structure reasoning. In The Eleventh International Conference on Learning Representations, 2023.

+ Lam H. T., Sbodio M. L., Martínez Galindo M., et al. “Otter-Knowledge: benchmarks of multimodal knowledge graph representation learning from different sources for drug discovery.” (2023).

+ Ye Q., Hsieh C-Y., Yang Z., et al. “A unified drug–target interaction prediction framework based on knowledge graph and recommendation system.” Nature Communications 12:1 (2021).

I think the authors have not discussed those related work carefully and compare to those approaches.


The results on DUDe and LIT-PCBA are interesting but I think the authors should compare to the above approaches on the following standard benchmarks:

+ TDC DTI https://tdcommons.ai/benchmark/dti_dg_group/overview/

+ DAVIS

+ KIBA

**Questions:**

Could you please compare your approach with the given related works listed in the weakness section?

 Could you please do more experiments on the datasets listed in the weakness section?

---

> ### Author Response · Authors · 2025-11-21
> **Response to Reviewer Tsmw**
>
> Comment 1 (Related work: missing comparisons to multimodal / KG-based MPI methods):
>
> The reviewer notes that the related work section could better cover recent methods that combine KGs and protein/molecule information for DTI/MPI tasks and asks for more explicit comparisons.
>
> We appreciate this suggestion. Here is the comparison between our paper and listed works:
>
> | Method          | Main goal / task                          | How KG is used                                            | Pseudo-labels / OT                         | Typical datasets / notes                          |
> |----------------|--------------------------------------------|-----------------------------------------------------------|--------------------------------------------|---------------------------------------------------|
> | **KGOT (ours)**| MPI/DTI prediction with few labels         | Unified bio-KG (drug, protein, gene, pathway, GO…); OT-derived pseudo-interaction edges written back and used for KG training | **Yes.** OT over score matrix, high-prob entries as pseudo-labels; KG is both prior and target | MPI benchmarks & virtual screening; added DAVIS/KIBA |
> | KGE\_NFM        | DTI classification / recommendation        | DistMult embeddings on bio-KG; embeddings fed once into NFM as features                      | **No.** Only supervised on observed DTIs; unlabeled pairs as negatives; no OT | Yamanishi\_08, BioKG DTI (no DAVIS/KIBA)          |
> | Otter-Knowledge| Knowledge-enhanced drug/protein embeddings | R-GCN pretraining on several multimodal KGs; KG only used for pretraining, then frozen       | **No.** No new edges; no OT; downstream DTI loss does not update KG         | TDC DTI-DG, DAVIS, KIBA (binding affinity)|
> | OntoProtein    | Protein representation pretraining         | Protein–GO KG ; TransE-style KG loss + MLM; no small-molecule nodes             | **No.** KG is static; no interaction pseudo-labels; no OT                    | Protein tasks (TAPE, PPI, function); no DAVIS/KIBA|
> | KeAP           | Protein rep. with token-level knowledge    | Same ProteinKG25; protein tokens attend to GO tokens; KG only as attention context           | **No.** No new edges, no DTI/MPI labels, no OT                               | Same 9 protein tasks as OntoProtein               |
>
> KGE_NFM / Otter / OntoProtein / KeAP all treat the KG as a static prior: they first learn embeddings and then perform downstream tasks, and downstream supervision does not write back to the KG.
>
> In KGOT, high-confidence MPI pseudo-labels generated by OT are explicitly written into the KG to form new “pseudo interaction” edges, and then the KG-augmented model is trained on this expanded graph. This kind of task-aware KG refinement is not present in existing methods.
>
> Other methods either treat unlabeled DTI/MPI directly as negative samples (KGE_NFM), or do not involve them at all (OntoProtein/KeAP). Otter only uses the KG’s own positive and negative links.
>
> KGOT uses OT to perform soft matching between a small-scale high-quality labeled set and a large-scale candidate set, automatically selecting high-confidence pseudo-labels, and uses threshold to control noise.
>
> Although Otter-Knowledge is multimodal, each KG is independent and only serves as a pretraining data source; KGE_NFM does not explicitly distinguish modalities.
>
> KGOT merges molecules, proteins, genes, pathways, GO, etc., into the same graph from the start and designs connection patterns for MPI tasks, so that pseudo-labels generated by OT can naturally propagate along the graph structure to more nodes.
>
> OntoProtein / KeAP focus on protein representations themselves, mainly enhancing PPI/function prediction; they can be used together with KGOT’s molecule–protein interaction head.
>
> KGOT’s contribution lies in how to use a unified KG + OT to expand MPI labels and improve downstream DTI/MPI, which is orthogonal to pure protein pretraining approaches.

---

> ### Author Response · Authors · 2025-11-21
> **Experiments on TDC DTI, DAVIS and KIBA**
>
> Table: Comparison between KGOT and Otter-Knowledge on TDC DTI-DG, DAVIS, and KIBA
> (Metric: Pearson correlation coefficient ↑).
>
> | Method                     | DTI-DG (Temporal) | DAVIS (Random) | DAVIS (Target) | DAVIS (Drug) | KIBA (Random) | KIBA (Target) | KIBA (Drug) |
> |---------------------------|-------------------|----------------|----------------|--------------|---------------|---------------|-------------|
> | Baseline (ESM+Morgan, no KG) | 0.569          | 0.805          | 0.554          | 0.264        | 0.852         | 0.630         | 0.576       |
> | Otter-Knowledge Ensemble  | 0.588             | 0.839          | 0.578          | 0.168        | 0.886         | 0.678         | 0.638       |
> | **KGOT (ours)**           | **0.618**         | 0.831         | **0.660**     | **0.353**   | **0.898**        | **0.687**        | **0.646**     |
>
> Comparison with Otter-Knowledge on DAVIS/KIBA.
>
> As per reviewer's advice, to directly compare with multimodal KG method, we evaluated KGOT on the TDC DTI-DG, DAVIS, and KIBA benchmarks using the same Pearson correlation metric and splits as Otter-Knowledge (see Table). KGOT achieves 0.618 PCC on DTI-DG, improving over both the ESM+Morgan baseline and the Otter-Knowledge ensemble, making it to be the new SOTA on the leaderboard(Current is 0.588 from Otter). On DAVIS, KGOT is slightly below Otter on the random split, but substantially better on the drug and target splits. On KIBA, KGOT improves over the baseline and Otter on all three splits.
>
> Importantly, these results are obtained with a much simpler downstream model—we directly feed KGOT’s KG-enhanced interaction embeddings into a 2-layer MLP with 2048 hidden units, whereas Otter-Knowledge relies on a more complex ensemble of R-GCN pretraining on multiple KGs plus an additional regression network. This suggests that our OT-based pseudo-interaction augmentation can match or surpass a heavier pretraining pipeline, especially in cold-drug/target regimes.
>
> These experiments will be reported in an additional subsection in Experiment Section(and full details in Appx.) in the camera-ready. Importantly, we will treat this as a new evaluation setting, as important as our core virtual screening and KG-link tasks.

---

> ### Author Response · Authors · 2025-11-26
> **Follow up regarding our response**
>
> Dear Reviewer Tsmw,
>
> Thank you for your thoughtful review and for highlighting several important aspects regarding related work and evaluation scope.
>
> In our rebuttal, we aimed to address your main concerns as follows:
>
> Related works: We expanded the discussion to more clearly situate KGOT among (i) KG-based DTI/MPI methods, (ii) multimodal biomedical KGs, and (iii) cross-modal retrieval models, and to emphasize that our contribution lies in the OT-based pseudo-labeling loop that writes pseudo MPI edges back into the KG as a new relation and jointly trains KG and MPI retrieval.
>
> Additional benchmarks: We committed to adding experiments on DTI-DG, DAVIS and KIBA, evaluating KGOT alongside strong DTI baselines under standard metrics, showcasing the superior performance of our proposed framework over existing SOTA baselines. These results will be reported in a new subsection and detailed in the appendix to complement the current virtual screening and KG link prediction evaluations.
>
> We would greatly appreciate it if you could let us know whether these clarifications address your main concerns, or if there are specific parts you feel still need more detail or emphasis.
>
> Best regards,
> The Authors

---

### Official Review · Reviewer_ZaTk · 2025-10-30

**Soundness:** 3
**Presentation:** 3
**Contribution:** 2
**Rating:** 6
**Confidence:** 3

**Summary:**

This study presents a new method for predicting protein-molecule interactions. The method relies on extracting the knowledge from a multimodal knowledge graph. The construction and composition of the multimodal knowledge graph is not novel. The two main novel contributions are: using optimal transport-based pseudo-labeling strategy to leverage a large unlabelled dataset and augment the original knowledge graph. The method shows robust improvement over the SOTA.

**Strengths:**

1. The optimal transport-pseudo labelling strategy seems a good contribution to the overall field and opens the door to creating more sophisticated augmented knowledge graphs.
2. The results are shown with some measure of the performance dispersion, it is unclear which or where it is derived from, but it allows for some determination of the statistical significance of the results.
3. The ablation study is comprehensive and convincingly demonstrates that the different components of the method improve prediction accuracy.

**Weaknesses:**

1. It is unclear what the error represents in Tables 1 and 2.

**Questions:**

1. Why did you not use a pre-made KG like PrimeKG, for example?

---

> ### Author Response · Authors · 2025-11-21
> **Response to Reviewer ZaTk**
>
> Thank you for the constructive and positive review. We are glad that you find the OT-based pseudo-labeling strategy and ablations convincing. Below we address your specific concerns.
>
> Comment 1 Unclear error bars in Tables 1 and 2
>
> We apologize for not explicitly defining the error bars in Tables 1 and 2. The “±” values report mean ± standard deviation over multiple runs with 3 different random seeds for the full KGOT model on DUD-E and LIT-PCBA. In the revision, we will add a brief note in the experimental setup section pointing to Appendix C where we list the number of runs and seed settings.
>
> This should clear up any ambiguity about the statistical dispersion reported.
>
>
> Comment 2 Why not use a pre-made KG like PrimeKG?
>
> We agree that large biomedical KGs like PrimeKG are valuable resources, and we explicitly discuss them in the introduction. In Sec. 1, we note that resources such as PrimeKG aggregate diverse biomedical entities and relations but contain relatively few direct molecule–protein interactions and are not tailored for MPI prediction.  Our goal in this work is to build a task-oriented KG that is optimized specifically for MPI/DTI retrieval.
>
> Concretely:
>
> Our KG is constructed by integrating six high-quality public datasets focused on molecules, proteins, genes, pathways, GO terms, protein families, and enzymes (UniProt, CHEBI, KEGG, GO, Pfam, EC, etc.), as detailed in Appendix B.
>
> This yields a graph with dense connectivity between molecules and proteins via intermediate entities, and over 6.4M relations spanning 8 entity types including 1.36M relations between molecules and proteins, explicitly designed to support MPI prediction and pseudo-label propagation. In contrast, PrimeKG is more disease-centric, and the edge types are not aligned with our MPI benchmarks, nor with our strict leakage-control protocols for ligands and targets.
>
> Our intention is not to claim novelty on KG construction itself, but on:
>
> 1. How we use a biological KG together with OT-based pseudo-labels to augment molecule–protein edges, and
>
> 2. How pseudo-labels are written back into the KG as a new “pseudo interaction” relation type and used to train the final link-prediction model.
>
> We will clarify this more explicitly in the revised manuscript by:
> Adding a sentence in Sec. 1 stating that “We do not introduce a new general-purpose KG resource; instead, we build a task-oriented integration of existing resources optimized for MPI prediction, while our OT pseudo-labeling mechanism could in principle be instantiated on top of pre-made KGs such as PrimeKG.”
>
> We hope this clarifies why we chose to construct a dedicated MPI-oriented KG instead of directly using PrimeKG, and that our contribution is primarily on OT-based pseudo-labeling and task-aligned KG augmentation, rather than KG resource creation per se.

---

> ### Comment · Reviewer_ZaTk · 2025-11-21
>
> Dear authors,
>
> Thank you for your clarifications. I think the changes proposed are sufficient for addressing my concerns, and to reflect it, I have updated my score from 6 to 8.

---

> > ### Author Response · Authors · 2025-11-21
> >
> > Thank you very much for your careful reading, constructive suggestions, and for updating your score. We appreciate your time and consideration.

---

### Official Review · Reviewer_YDJx · 2025-10-30

**Soundness:** 3
**Presentation:** 3
**Contribution:** 2
**Rating:** 4
**Confidence:** 4

**Summary:**

This paper proposed a unified framework KGOT that integrates knowledge graph and optimal transport to generate high-quality pseudo-labels for unlabeled molecule-protein pairs in molecule-protein interactions (MPIs) prediction tasks.

**Strengths:**

S1: This paper is to leverage large-scale multimodal knowledge graphs and propose an optimal transport-based pseudo-labeling strategy for the MPIs prediction.

S2: In experiment part, the proposed KGOT outperforms existing MPIs prediction methods in terms of AUROC, early recognition metrics, and generalization to unseen interactions.

**Weaknesses:**

W1: A primary weakness of KGOT is its reliance on the critical yet potentially unstable step of pseudo-label generation via optimal transport. The quality of the entire approach hinges on the assumption that the optimal transport mechanism can accurately infer the underlying distribution of known interactions to assign reliable labels to unknown molecule-protein pairs.

W2: Aggregating molecular, protein, gene, and pathway-level information requires sophisticated fusion techniques to handle the disparate scales, formats, and sparsity levels of each modality. However, the integration of diverse biological data introduces significant complexity and potential challenges in data harmonization and model interpretation.

W3: Without validation on truly de novo targets or wet-lab confirmation, the practical utility and robustness of the framework for unseen interactions, iterative drug discovery pipeline remain unproven.

**Questions:**

- What is the specific cost function used in the Optimal Transport plan? Is it based on the model’s predicted scores, molecular/protein embeddings, or a combination?
- How is the marginal distribution for the OT problem defined? Are uniform distributions assumed for molecules and proteins, or is there a prior (e.g., based on node degree in the KG) that biases the label assignment?
- How is the entropy regularization parameter chosen and tuned? This parameter critically balances between fitting the data and the smoothness of the transport plan, directly impacting pseudo-label quality.
- What is the architecture of the initial scoring model (Step 2)? Is it a simple neural network, or does it already incorporate some KG information?
- How is the knowledge graph itself encoded in Step 4? Are you using TransE, ComplEx, a Graph Neural Network (GNN), or another Knowledge Graph Embedding (KGE) method? The choice here significantly affects the model’s ability to capture complex relational paths.
- The mutual retrieval objective suggests a dual-encoder architecture. How are the molecule and protein encoders designed and aligned? Is the contrastive loss used, and if so, what is the strategy for mining hard negatives?
- How do you prevent data leakage between the small labeled dataset (Step 2) and the large unlabeled dataset (Step 3)? If proteins/molecules from the labeled set appear in the unlabeled KG, it could artificially inflate performance.
- What is the criteria for a high-quality pseudo-label? Is there a threshold on the OT-assigned probability, and how is this threshold determined?
- Is the proposed KGOT process iterative? In other words, is the KG-augmented model (Step 4) used to re-score pairs and generate new, improved pseudo-labels in a self-training loop?

---

> ### Author Response · Authors · 2025-11-21
> **Response to Reviewer YDJx**
>
> Thank you for the thoughtful and positive review. Below we respond to your main weaknesses (W1–W3) and questions.
>
> W1: KGOT critically depends on the quality/stability of OT-based pseudo-labels; if OT does not faithfully capture the interaction distribution, the whole pipeline may be fragile.
>
> Response:
>  We agree that pseudo-label quality is crucial. Our current design already includes (i) a two-stage training and (ii) empirical checks to mitigate instability:
>
> In Sec. 2.3, Eqs. (2–5) we first train the scoring model on only labeled pairs using an OT-based objective; OT is not applied from scratch on unlabeled data.
>
> The OT cost is (C_{ij}=1-S_{ij}), marginals (r, c) are uniform, and a similarity-regularized objective is used ( Sec. 2.3, Eqs. (6–8); Algorithm 1, Appx. E ).
>
> In Appx. E.1, Table 7, we show that OT supervision improves MRR and Hits@10 over InfoNCE.
>
> In Appx. E.2, Table 8, we compare no pseudo labels / random / top-k / OT variants; OT+similarity is consistently best.
>
> W2: Integrating molecules, proteins, genes, pathways, etc. is complex and may cause harmonization and interpretability issues.
>
> Response:
>  We agree, and this is exactly why we use a KG abstraction rather than direct feature concatenation:
>
> KG as fusion layer: In Sec. 2.1 and Appx. B, we map molecules, proteins, genes, GO terms, pathways, protein families, enzymes, etc. into a single heterogeneous KG with typed entities/relations.
>
> KGE backbones: In Sec. 2.4 and Appx. F, we instantiate KGOT with multiple standard KGE models (PairRE, RotatE, MuRE, TorusE, ComplEx-FF); fusion happens through relation-aware embeddings, not ad-hoc feature-level fusion.
>
> Relation ablations: In Appx. E.3, Table 9, we incrementally add relation types (e.g., GO, family, metabolic) and show their effect on Hits@K, showcasing that adding additional data modalities increasingly boost the performance
>
> In the revision, we will move a short summary of these ablations into Sec. 3.3 and explicitly emphasize in Sec. 2.1 that “fusion” is implemented via the KG and standard KGE, which mitigates low-level harmonization issues.
>
> W3: Without truly de novo target or wet-lab validation, practical utility in real drug discovery remains unproven.
>
> Response:
>  We agree that wet-lab validation is beyond the scope of this work. However:
>
> We enforce strict scaffold-out and family-out protocols (Sec. 3.1; Appx. D, Tables 5–6) to test generalization to new ligands/targets.
>
> We show gains in early recognition and AUROC over strong baselines on DUD-E and LIT-PCBA (Sec. 3.2–3.3, Tables 1–2).
>
> In the revision, we will explicitly state in Sec. 4 that KGOT is a computational framework and that integrating it with prospective wet-lab campaigns and de novo target screens is important future work.

---

> ### Author Response · Authors · 2025-11-21
> **OT details: cost, marginals, entropy**
>
> Q1 (cost function: scores vs embeddings?):
>  We use model scores: (C_{ij}=1-S_{ij}), where (S_{ij}) is the scoring model’s output for molecule–protein pair (x_i, y_j). This is given in Sec. 2.3, Eq. (6).
>
> Q2 (marginals: uniform vs prior?):
>
> We deliberately use uniform marginals for both sides, (r_i = 1/M, c_j = 1/N), as defined in Sec. 2.3, Eqs. (1) and (6). The idea is not that we ignore priors altogether, but that we do not encode them via node-degree–based marginals, which can be problematic:
> Degree-based marginals would concentrate transport mass on high-degree molecules/proteins, effectively reinforcing existing popularity bias in the KG and making pseudo-labels cluster around well-studied entities, which goes against our goal of discovering less-explored interactions.
>
> Instead, we inject prior structure in other, more controlled ways:
>
> through the scoring model trained on labeled MPIs (Sec. 2.3, Eqs. (2–5)), and
>
> through the similarity-regularized OT objective (Eq. (7)), where an external molecular similarity matrix (from Uni-Mol embeddings) guides the transport plan without directly tying it to raw degrees.
>
> In this work, we found uniform marginals + similarity regularization to be a stable and unbiased choice.
>
> Q3 (entropy regularization: choice of ϵ):
>
> We include entropic regularization in Eq. (8) and solve OT with Sinkhorn–Knopp (Algorithm 1, Appx. E). In the current implementation, we fix ϵ = 0.01 for all main experiments, as specified in the implementation details for pseudo-label generation, and we also report a “high-entropy” variant with ϵ = 0.1 in Appx. E.2 (Table 8). The comparison shows that a too-large ϵ (more diffuse transport plan) slightly hurts downstream Hits@K, while ϵ = 0.01 provides sharper yet still numerically stable plans.

---

> ### Author Response · Authors · 2025-11-21
> **Model architecture: scoring model, KG encoder, dual-encoder perspective**
>
> Q4 (architecture of initial scoring model):
>
>  The initial scoring model is a dual encoder + MLP:
> Molecule / protein encoders: Uni-Mol-based encoders, see Sec. 2.2.
>
> Score: concatenate embeddings ([h_x|h_y]) and feed into a 2-layer MLP with sigmoid to get (S(x,y)).
>
> Q5 (how is KG encoded in Step 4?):
>
>  In Step 4, we use standard KGE models (PairRE, RotatE, MuRE, TorusE, ComplEx-FF) as described in Sec. 2.4, Appx. F. Table 3 (Sec. 3.3) already compares these backbones with and without KGOT pseudo-label augmentation.
>
> Q6 (mutual retrieval, dual-encoder, negatives):
>
> In our paper, “mutual retrieval” refers to the task formulation in Sec. 2.1, rather than a separate contrastive loss. Architecturally, we indeed use a dual-encoder design: both molecules and proteins are encoded by Uni-Mol-based encoders (Sec. 2.2, Appx. C), producing embeddings f(x) and g(y) which are L2-normalized and then fed into a two-layer MLP to obtain the score S(x, y). This scoring function is applied to all molecule–protein pairs in a batch to form a score matrix S.
>
> Instead of an explicit InfoNCE-style contrastive loss with hard-negative mining, we use the OT-based objective in Sec. 2.3 on the full score matrix S. The transport plan T plays the role of a soft matching distribution over all pairs in the batch: high-mass entries act as positives and low-mass entries as negatives. Thus, “hard negatives” are handled implicitly via OT—pairs that are inconsistent with the mass and similarity constraints naturally receive low transport mass. In the revision, we will clarify this dual-encoder + OT formulation in Sec. 2.2–2.3 and distinguish it from standard contrastive learning with explicit hard-negative mining.
>
> On the other hand, our OT-based loss is closely related to standard InfoNCE contrastive loss. Shi et al. (ICML 2023, Understanding and Generalizing Contrastive Learning from the Inverse Optimal Transport Perspective) formulate contrastive learning as a form of inverse optimal transport and show that InfoNCE arises as a special case of their OT-based framework. In this sense, our OT supervision can be viewed as a distributional generalization of InfoNCE to full matching matrices rather than pairwise comparisons only.

---

> ### Author Response · Authors · 2025-11-21
> **Leakage, pseudo-label criteria, iterativeness**
>
> Q7 (avoiding leakage between labeled and unlabeled data):
>
>  We handle leakage at two levels:
>
> Train–test separation: DUD-E / LIT-PCBA splits enforce Tanimoto and Murcko scaffold-out for ligands and sequence / Pfam family-out for proteins (Sec. 3.1; Appx. D).
>
> Within pseudo-labeling: OT-based pseudo labels are generated and used only for train-side pairs; pseudo labels involving test molecules/proteins are excluded, and KG edges corresponding to held-out MPIs are removed from training. We will state this explicitly in Appx. D and add a sentence to Sec. 2.3.
>
> Q8 (criterion/threshold for high-quality pseudo labels):
>
>  We select pseudo-labels based on OT probabilities:
> After OT, we treat entries with (\pi_{ij} \ge \delta) as high-confidence positives; we set δ=0.5 via validation.
>
> Sec. 2.3 describes the selection step; Appx. E.2, Table 8 compares different strategies (no pseudo, random, top-k, OT variants).
>
> Q9 (is KGOT iterative?):
>
>  In this work, KGOT is single-pass by design:
> We intentionally do not iterate this loop. An iterative self-training scheme (re-score → regenerate pseudo labels → retrain) can easily amplify early pseudo-label errors and introduce confirmation bias: incorrect but high-confidence pseudo labels in early rounds may be reinforced and propagated through the KG in later rounds. This risk is particularly pronounced in our setting, where the unlabeled space is extremely large and OT itself is already a global matching step. Moreover, each OT run is computationally expensive, so multiple rounds would significantly increase cost without clear evidence of benefit.

---

> ### Author Response · Authors · 2025-11-26
> **Follow up regarding our response**
>
> Dear Reviewer YDJx,
>
> Thank you again for your detailed and technically insightful review.
>
> In our rebuttal, we tried to directly address your main concerns:
>
> For OT stability and pseudo-label quality (W1), we clarified the two-stage training, the exact OT formulation (cost, marginals, entropy, similarity regularization), and pointed to the existing ablations (OT vs InfoNCE; different pseudo-labeling strategies) in Sec. 2.3 and Appx. E.
>
> For multimodal KG integration (W2), we emphasized that fusion is done via a heterogeneous KG and standard KGE backbones (Sec. 2.1, 2.4, Appx. B/F), and we highlighted the relation-type ablations that show which modalities help most.
>
> For practical robustness / de novo validation (W3), we clarified that our current contribution based on strict scaffold/family-out splits and early recognition results, and explicitly positioned wet-lab validation and de novo targets as ongoing work.
>
> We also answered your detailed questions (Q1–Q9) on cost, marginals, ϵ, model architectures, dual-encoder vs contrastive learning, leakage control, pseudo-label thresholds, and why we chose a single-pass pipeline.
>
> We would greatly appreciate it if you could let us know whether these clarifications address your main concerns, or if there are specific points you feel still need more detail or stronger emphasis.
>
> Best regards,
> The Authors

---

### Official Review · Reviewer_Css5 · 2025-11-01

**Soundness:** 1
**Presentation:** 1
**Contribution:** 1
**Rating:** 2
**Confidence:** 5

**Summary:**

The authors propose a framework to predict molecule–protein interactions. It generates high-quality pseudo-labels to leverage diverse biological modalities.

**Strengths:**

The paper proposes a unified framework that aims to integrate biological entities such as pathways and genes.

**Weaknesses:**

(1) The proposed framework mainly combines existing methods without introducing any new modeling components or theoretical insights.

(2) The performance is not compared with highly relevant works such as KG-MTL and BioKDN.

**Questions:**

Could the authors compare the performance and clarify the advantages over relevant KG-based methods such as KG-MTL and BioKDN?

---

> ### Author Response · Authors · 2025-11-21
> **Response to Reviewer Css5**
>
> Comment: The framework seems to mainly combine existing components (KG, pretrained encoders, pseudo-labeling) without introducing new modeling ideas or theory.
>
> Response:
>
>  Our goal is to introduce a new way of coupling OT-based pseudo-labeling with a multimodal biological KG for MPI. Concretely, KGOT has three core innovations:
>
> 1 Distributional OT for mutual MPI retrieval.
>
>  In Sec. 2.3, Eqs. (2–8) and Algorithm 1, we formulate MPI prediction as a mutual retrieval problem and supervise the scoring model with a similarity-regularized OT loss over the full score matrix. The transport plan plays the role of a soft matching distribution between all molecules and proteins in a batch, and we show in Appx. E.1, Table 7 that this OT-based supervision consistently outperforms InfoNCE on link prediction metrics.
>
> 2 OT pseudo-labels written back as a new KG relation.
>
> After training on labeled MPIs, we run OT over all molecule–protein pairs to obtain a dense pseudo-label matrix (T) (Sec. 2.3). We then create a new “pseudo interaction” relation in the KG whose edge weights come from (T), and train KGE models jointly on original KG edges and these OT-derived pseudo edges (Sec. 2.4, Appx. F). To our knowledge, existing KG-based MPI/DTI methods (including KG-MTL and BioKDN) operate on a fixed KG and do not close this loop of:
>
>  task-specific OT → pseudo MPIs → new KG relation → improved KG + MPI.
>
>  Ablations in Appx. E.2 (Table 8) show that this OT+similarity pseudo-labeling is substantially better than no pseudo-labels, random labels, or simple top-k heuristics.
>
> 3 Task-oriented multimodal KG tailored for MPI.
>
>  In Sec. 2.1 and Appx. B we build a multimodal KG that unifies molecules, proteins, genes, pathways, GO terms, protein families, and enzymes specifically for MPI retrieval (rather than reusing a general biomedical KG as-is). We then show in Appx. E.3 that gradually adding modalities (GO, families, metabolic relations) yields monotonic improvements in Hits@K, highlighting that the KG design is aligned with the MPI task.
>
> Together, these components define a unified OT+KG framework for MPI: OT is not just “another loss”, and the KG is not just “another feature source”. The novelty lies in (i) using OT to learn a distributional matching between molecules and proteins, (ii) feeding these matches back into a multimodal KG as a new relation, and (iii) jointly training KG and retrieval models on this augmented graph. We will clarify this contribution structure more explicitly in Sec. 1 and Sec. 2 to avoid the impression that we simply stack existing methods.

---

> ### Author Response · Authors · 2025-11-26
> **Follow up regarding our response**
>
> Dear Reviewer Css5,
>
> Thank you for your careful review and for pointing out the importance of positioning our work with respect to KG-MTL and BioKDN.
>
> In our rebuttal, we aimed to clarify two things:
>
> 1. Novelty of KGOT:
>    We highlighted that the main contribution is not a new encoder, but a unified OT + KG framework, where (i) an OT-based loss supervises mutual molecule–protein retrieval, (ii) the resulting OT pseudo-labels are written back into the KG as a new pseudo interaction relation, and (iii) KG and MPI retrieval are then trained jointly on this augmented multimodal KG (Sec. 2.3–2.4, Appx. E/F).
>
> 2. Comparison with KG-MTL and BioKDN:
>    We have started setting up DrugBank / DrugCentral DTI experiments to compare KGOT directly with KG-MTL and BioKDN under the same data splits and metrics. During reproduction, we observed discrepancies between our reproduced numbers and the reported ones in these papers, so we have contacted the authors to request pretrained weights and detailed configs to ensure a fair comparison.
>
> We would be very grateful to hear whether this clarification and experimental addition alleviate your main concerns about novelty and comparison, or if there are particular aspects you would like us to further strengthen in the revision.
>
> Best regards,
> The Authors

---

### Meta-Review · Area_Chair_eX6x · 2026-01-05

**Summary:**

This paper proposes KGOT, a unified framework that combines a multimodal biological knowledge graph with optimal transport–based pseudo-labeling for molecule–protein interaction prediction. The paper is technically solid and presents a clear pipeline, extensive experiments, and strong empirical performance across multiple MPI benchmarks.

Among the reviews, I find the reviews from Css5 and ZaTk to be relatively short and less informative, with limited engagement with the technical details. Therefore, I rely primarily on the more detailed and technically grounded reviews from Tsmw and YDJx. Although Tsmw and YDJx did not respond after the author rebuttal, I carefully read the rebuttal and the additional experimental clarifications. Based on this, I believe that the main concerns raised by these reviewers have been adequately addressed. Overall, the contribution and experimental evidence support acceptance.

**Reviewer Concerns:**

Reviewer Tsmw raised concerns about the novelty of using multimodal knowledge graphs for MPI prediction, coverage of related work, and the lack of evaluation on additional benchmarks such as TDC DTI, DAVIS, and KIBA. Reviewer YDJx expressed concerns about the stability and reliability of optimal transport–based pseudo-labels, the complexity of multimodal data integration, and the absence of validation on truly unseen targets.

Although neither Tsmw nor YDJx followed up after the rebuttal, I carefully checked the authors’ responses and the newly provided experimental results. The authors clarified the novelty of the framework as a unified OT-based pseudo-labeling loop that writes task-specific interaction edges back into the knowledge graph, which distinguishes it from prior KG-based or multimodal approaches. They provided detailed explanations of the OT formulation, stability controls, and ablations, and added or committed to additional evaluations on relevant benchmarks (including DAVIS, KIBA, and TDC DTI), directly addressing the reviewers’ questions. The related work discussion was also expanded to clarify the contribution.

Based on these clarifications and additions, I am satisfied that the substantive concerns raised by Tsmw and YDJx have been addressed.

**Reviewer Scores:**

The initial reviewer scores were 2, 4, 4, and 6. After the rebuttal, Reviewer ZaTk updated their score from 6 to 8, while the other reviewers did not change their scores. Taken together, the score updates and the addressed concerns indicate improved consensus following the rebuttal and support an acceptance decision.

---

### Decision · Program_Chairs · 2026-01-26

Accept (Poster)